METHODS

# aiSEGcell: User-friendly deep learning-based segmentation of nuclei in transmitted light images

Daniel Schirmacher ⓘ*, Ümmünur Armagan ⓘ, Yang Zhang, Tobias Kull, Markus Auler, Timm Schroeder ⓘ*

Department of Biosystems Science and Engineering, ETH Zurich, Basel, Switzerland

* daniel.schirmacher@bsse.ethz.ch (DS); timm.schroeder@bsse.ethz.ch (TS)

## Abstract

Segmentation is required to quantify cellular structures in microscopic images. This typically requires their fluorescent labeling. Convolutional neural networks (CNNs) can detect these structures also in only transmitted light images. This eliminates the need for transgenic or dye fluorescent labeling, frees up imaging channels, reduces phototoxicity and speeds up imaging. However, this approach currently requires optimized experimental conditions and computational specialists. Here, we introduce "aiSEGcell" a user-friendly CNN-based software to segment nuclei and cells in bright field images. We extensively evaluated it for nucleus segmentation in different primary cell types in 2D cultures from different imaging modalities in hand-curated published and novel imaging data sets. We provide this curated ground-truth data with 1.1 million nuclei in 20,000 images. aiSEGcell accurately segments nuclei from even challenging bright field images, very similar to manual segmentation. It retains biologically relevant information, e.g. for demanding quantification of noisy biosensors reporting signaling pathway activity dynamics. aiSEGcell is readily adaptable to new use cases with only 32 images required for retraining. aiSEGcell is accessible through both a command line, and a napari graphical user interface. It is agnostic to computational environments and does not require user expert coding experience.

**Data Availability Statement:** Input data for all analysis scripts, data sets D1-D5, analysis scripts used for all figures, and paper versions of command line interface and graphical user

## Author summary

Fluorescence microscopy is the most widely used method to monitor cellular structures in space and time. Fluorescently labeling cellular structures is typically required to localize ("segment") them in electronic images for subsequent quantification. Deep learning approaches can detect these structures also in only bright field images. This eliminates the need for a fluorescent label, frees up imaging channels, speeds up imaging, and reduces the harmful effects of exposing cells to high intensity light. However, label free segmentation currently requires optimized experimental conditions and computational specialists. Therefore, we developed "aiSEGcell" a user-friendly deep learning-based software to segment nuclei and cells in only bright field images. We extensively evaluated aiSEGcell on different common experimental conditions and showed that biologically even sensitive

interface code are available at the ETH Research Collection repository https://doi.org/10.3929/ethz-b-000679085. The actively maintained aiSEGcell command line interface code is available at https://github.com/CSDGroup/aisegcell, and the aiSEGcell napari plugin code is available at https://github.com/CSDGroup/napari-aisegcell.

**Funding:** This project was funded in part by a grant (2022-309751) from Chan Zuckerberg Initiative DAF (https://chanzuckerberg.com) an advised fund of Silicon Valley Community Foundation to D.S. and T.S. and Swiss National Science Foundation (https://www.snf.ch) grant (186271) to T.S.. D.S. received a salary from grant 2022-309751 and 186271. Y.Z. received a salary from grant 186271. The funders had no role in study design, data collection and analysis, decision to publish, or preparation of the manuscript.

**Competing interests:** The authors have declared that no competing interests exist.

relevant information is retained. Furthermore, we demonstrated that aiSEGcell is adaptable by retraining to new applications with very little required data. We make it accessible for users with no required expert coding experience in a wide range of computational environments. Finally, we openly share our very large imaging data sets to further the development of other segmentation approaches.

## Introduction

Image segmentation—the partitioning of an image into meaningful segments—is fundamental to image analysis. It is usually the first step in image analysis pipelines to identify tissues, cells or subcellular structures to then allow their state quantification or tracking and dynamics quantification [1]. Accurate segmentation is usually crucial for robust automated downstream analyses [2]. While manual image segmentation is considered the gold standard, segmentation for high throughput microscopy requires computer assistance. Conventional computer-assisted image segmentation is semi-automated and necessitates good signal-to-noise, using a dedicated fluorescence channel for segmenting cells, nuclei, or other targets [3–5]. However, dedicating a fluorescence channel to facilitate image segmentation loses a channel that would otherwise be available to image other relevant targets and increases imaging phototoxicity and required acquisition times [6]. Furthermore, adding a transgenic fluorescence marker increases the size of genetic constructs, complicates experimentation and reduces virus packaging and transduction efficiency [7]. Similarly, the time and cost associated with establishing and maintaining a transgenic animal line increases considerably with each additional transgene. Eliminating the need for fluorescent markers for segmenting cellular structures would thus greatly improve imaging experiments.

This is made possible by segmenting cells and their compartments in transmitted light images. Albeit visible in transmitted light images, features indicative of even a large organelle like the nucleus are complex, vary between cell types, and over time, for example due to cell differentiation, cell movement or changes in the focal plane. Neural networks excel at discovering complex informative features and are not limited to specific imaging conditions or cell types if trained on a large representative data set [8–11]. Previous research has shown that neural networks can automatically detect nuclei and other sub-cellular structures in high magnification bright field (BF) images [12,13]. Moreover, U-Net has been suggested as a well-suited model architecture for this purpose, potentially even for low magnification imaging [14–16]. However, it is uncertain whether nuclei can be accurately segmented under challenging typical imaging conditions optimized for speed and cells' health and not for perfect images, such as low magnification BF microscopy. Furthermore, actively maintained, user-friendly software for segmenting nuclei in BF is not available. Consequently, widespread experimental adoption remains elusive.

Here, we present "aiSEGcell" a user-friendly software to segment nuclei in BF images accessible even to users with no prior coding experience. We demonstrate its utility even in challenging prevalent conditions, including low magnification live time lapse imaging of primary cells. This includes small hematopoietic stem and progenitor cells lacking prominent and large cell compartments like cytoplasm, also in microfluidic and other culture devices. More specifically, we quantify both technical performance metrics and the preservation of biologically meaningful information such as live single cell signaling dynamics. We show that our trained instance of aiSEGcell is transferable to unseen experimental conditions and adaptable to other use cases, like novel cell types, only requiring 32 images for retraining.

Finally, we demonstrate aiSEGcell's benefits by quantifying an additional fluorescence channel in an experimental setting where the number of biomarkers is limited by their spectral properties. aiSEGcell is accessible through a command line interface (CLI) or a graphical user interface (GUI), a napari plugin, tested by users with varying coding experience in different computational environments [17]. In addition, we provide the five used data sets, containing more than 20,000 images, and 1,100,000 nuclei, as open access data through a FAIR repository [18].

## Results

### Segmenting nuclei in common bright field images

Conventional cell segmentation approaches use fluorescence to mark cells and cell compartments for segmentation (Fig 1a). In contrast, aiSEGcell omits fluorescence markers for segmentation and directly segments BF images (Fig 1b). We selected U-Net as the model architecture [12,14] for aiSEGcell and trained it on 9,849 BF image and segmentation mask pairs (>600,000 nuclei) of live primary murine hematopoietic cells, like macrophages, stem and progenitor cells, at 20x magnification (D1) [16]. To highlight the need for user-friendly software that segments nuclei in BF, we compared aiSEGcell to widely used nucleus segmentation software Cellpose and StarDist [19,20]. Available pretrained models for nucleus and BF segmentations were not suitable alternatives to the specialized BF nucleus segmentation of aiSEGcell (S1 Fig and S1 and S2 Tables). We evaluated aiSEGcell on a second data set (D2) comprised of 6,243 image-mask pairs (>300,000 nuclei) derived from five independent imaging experiments with experimental settings similar to D1. Cell morphologies included round

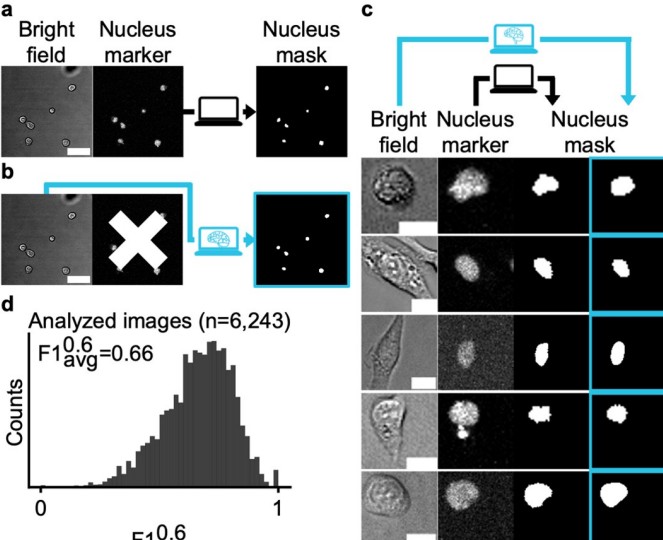

**Fig 1. aiSEGcell accurately segments nuclei in bright field images.** (a) Workflow to segment nuclei based on nuclear marker or (b) BF (cyan; scale bars 40 μm). (c) Nuclear segmentations in BF were qualitatively similar to fluorescent nuclear marker derived segmentations. Each field-of-view stems from a different independent experiment of data set D2 (scale bars 10 μm). (d) Nuclear segmentations in BF (b) were quantitatively similar to nuclear marker derived segmentations (a) in D2 (n = 6,243 images, N = 5 experiments). The F1-score ranges between 1 (best) and 0 (worst) and the image-wise average F1 with an intersection over union threshold of 0.6 is denoted as $F1_{avg}^{0.6}$. Given the challenges of imaging live primary non-adherent cells we consider $F1^{0.6} \geq 0.6$ good. The results in Figs 2, 3c and 4c support this estimate. Images were cropped from a larger field-of-view and contrast was individually adjusted here for visibility.

**Table 1. Data sets overviews.**

| Data set | D1 | D2 | D3 | D4 | D5 |
|---|---|---|---|---|---|
| Cell types | mGMP, mMPP, mPreGM, mPreMegE, mMkP, mHSC, cMoP, mMac | mPreGM, mMkP, mPreMegE, mMac, cMoP, mGMP | mHSC, mMEG | mESC | hMPP |
| Magnification | 20x | 20x | 20x | 10x | 20x |
| Microscopes | 2 | 2 | 2 | 2 | 1 |
| Experimenters | 3 | 4 | 1 | 1 | 1 |
| Images train | 8,864 | - | 24 | 22 | - |
| Nuclei train | 554,228 | - | 2,702 | 2,790 | - |
| Exp train | 14 | - | 1 | 1 | - |
| Images validation | 985 | - | 10 | 10 | - |
| Nuclei validation | 66,894 | - | 1,381 | 1,025 | - |
| Exp validation | 14 | - | 1 | 1 | - |
| Images test | 3,153 | 6,243 | 29 | 10 | 816 |
| Nuclei test | 193,973 | 311,091 | 979 | 2,712 | 9,556 |
| Exp test | 12 | 5 | 1 | 1 | 1 |
| Exp total | 14 | 5 | 3 | 3 | 1 |
| Visual artifacts | - | - | - | - | microfluidic chip |

List of abbreviations: independent experiment (Exp), murine (m), human (h), granulocyte/monocyte progenitor (GMP), pre-granulocyte/monocyte progenitors (PreGM), pre-megakaryocytic/erythroid progenitor (PreMegE), megakaryocyte progenitors (MkP), hematopoietic stem cell (HSC), common monocytic progenitor (cMoP), macrophage (Mac), megakaryocyte (MEG), embryonic stem cell (ESC), multipotent progenitor (MPP).

cells and more complex shaped cells with protruding pseudopodia. In addition, cell sizes varied by an order of magnitude (Table 1). aiSEGcell segmented nuclei in BF images across morphologically distinct cells and different scales (Figs 1c and S2a). We quantified segmentation quality using an adapted version of the F1-score that counted inaccurately predicted objects (e.g. intersection over union = 0.4) as one error instead of as both, false positive (FP), and as false negative (FN; see Materials and methods). Therefore, the image-wise average adapted F1-score with an intersection over union (IOU) threshold of 0.6, denoted as $F1_{avg}^{0.6}$, more appropriately represented the observed segmentation quality [21]. aiSEGcell nuclear segmentations in BF were quantitatively similar to nuclear marker derived segmentations (ground truth; $F1_{avg}^{0.6} = 0.66$). Around 70% versus 5% of images had $F1^{0.6} \geq 0.6$ versus $F1^{0.6} \leq 0.4$, illustrating that aiSEGcell segmentations are consistently accurate across challenging typical BF images (Fig 1d and S3 Table). Furthermore, we quantified the frequency of oversegmentations (an object like a nucleus in the ground truth segmentation is split into multiple objects in the aiSEGcell segmentation) and undersegmentations (multiple objects in the ground truth segmentation are merged into a single object in the aiSEGcell segmentation). Only 0.2 (s.d. = 0.5) of 50 ground truth nuclei were oversegmented in aiSEGcell segmentations and 0.2 (s.d. = 0.6) of 50 nuclei detected in aiSEGcell were merged ground truth nuclei per image on average in D2 (S4 Table). Images in D2 were intended for cell tracking and consequently not of high cell density. To evaluate if undersegmentations were more frequent in images dense with cells, we acquired an additional data set (D7) with densely seeded cells. aiSEGcell accurately ($F1_{avg}^{0.6} = 0.80$) segmented nuclei in BF and undersegmentations rarely occurred. Only 0.2

(s.d. = 0.5) out of 50 nuclei detected by aiSEGcell were merged ground truth nuclei per image on average (S3 Fig and S4 and S5 Tables).

## aiSEGcell is sufficient to quantify noisy dynamics data

Mask-level performance evaluation of segmentation models may not be relevant to illustrate suitability for downstream applications. Consequently, evaluating segmentation models with respect to biologically meaningful read-outs is necessary. The translocation of fluorescently labeled signaling molecules from the cytoplasm to the nucleus and vice versa, in small motile live primary cells imaged with low signal-to-noise is demanding, sensitive to variations in the nuclear mask and has been shown to affect cell behavior and fate [22–24]. We used a homozygous reporter mouse-line that had all p65 proteins, the transcriptionally active part of the nuclear factor κB (NfκB) dimer, tagged with green fluorescent protein (GFP) to quantify its translocation to the nucleus upon NfκB signaling activation (Fig 2a). We imaged highly motile live primary macrophages, hematopoietic stem and progenitor cells in conditions optimized for cell health and acquisition speed resulting in low signal-to-noise images—a realistic challenging experimental setting that tests the feasibility of aiSEGcell segmentations for many other applications. Previous research showed that hematopoietic cells stimulated with tumor necrosis factor α (TNFα) respond with either of four characteristic NfκB signaling dynamics response types: non-responsive (NON), oscillatory (OSC), sustained (SUS), or transient (TRA; Fig 2b) [25]. We tracked 458 single cells exposed to control media or TNFα for up to 23 h in the five independent time lapse experiments of D2 (Table 1). With processing and quantification otherwise identical, we compared the NfκB responses derived from aiSEGcell (trained on D1) and ground truth segmentations. We observed that single cell signaling dynamics classifications based on aiSEGcell segmentations closely matched those based on ground truth segmentations (average Euclidean distance = 0.53, n = 458; Fig 2b). Moreover, 81% of aiSEGcell signaling dynamics were assigned the same response type as the corresponding signaling dynamics obtained from ground truth segmentations. Classifications were accurate for all response types (NON: 88%, OSC: 77%, SUS: 79%, TRA: 74% match) and for the different experiments contributing to D2 (Figs 2c and S4). Consequently, aiSEGcell segmentations do not introduce a bias towards NfκB response types and retain the characteristic response type frequencies of different primary hematopoietic cells.

## aiSEGcell is easily adaptable to new experimental conditions

Once trained, neural networks have the potential to be applied indefinitely to data of the same distribution, in some cases even to new data distributions. This training process is notoriously challenging and requires hardware resources that are prohibitive for many users. Consequently, it is important to investigate the robustness of a neural network to unseen data. We tested aiSEGcell trained on D1 on a published unseen data set (D5) comprised of human multipotent progenitor populations (hMPP) cultured in a microfluidic chip acquired by an unseen experimenter on an unseen microscopy setup (Figs 3a and S2b and Tables 1 and S6). We repeated the published data processing to obtain single cell extracellular signal-regulated kinase (ERK) pathway signaling dynamics, now using aiSEGcell segmentation masks. Finally, we automatically assigned ERK signaling dynamics to one of four response types: NON, TRA, intermediate (INT), SUS [22,26]. ERK signaling dynamics obtained from aiSEGcell segmentations were similar to those based on ground truth segmentations (average Euclidean distance = 0.50, n = 126; Fig 3b). Overall, the signaling dynamics classifications corresponded very well between manually curated or aiSEGcell segmented nuclei, illustrating aiSEGcell's usability: 81% of aiSEGcell response type assignments matched the ground truth signaling

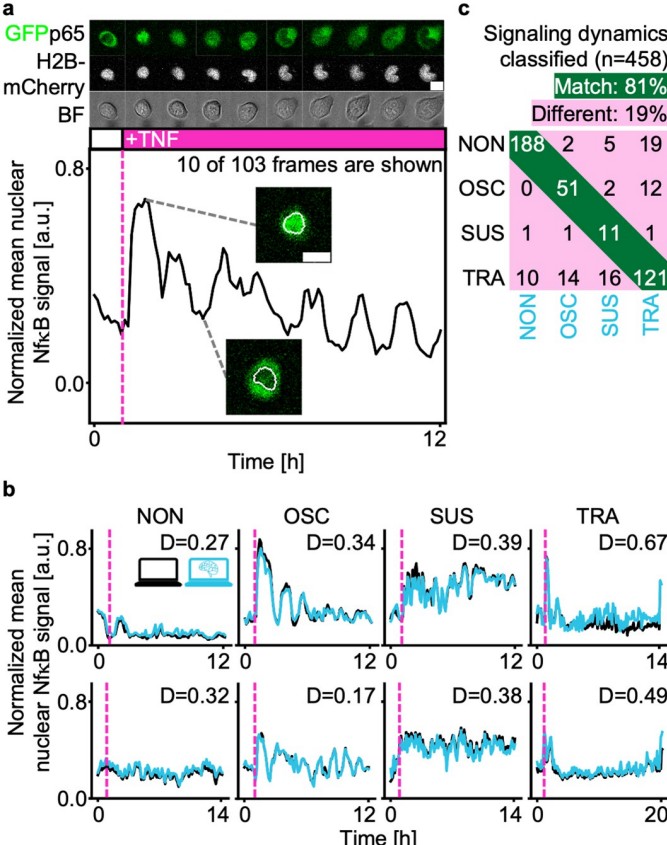

**Fig 2. aiSEGcell is sufficient to retain demanding biologically meaningful information.** (a) NfκB signaling dynamics obtained from a representative single tracked primary murine granulocyte/monocyte progenitor cell. The magenta bar and dashed line illustrate a single addition of TNFα after 1 h. Contrast was individually adjusted here to improve visibility (scale bars 10 μm). (b) Signaling dynamics in D2 divide into four response types: non-responsive (NON), oscillatory (OSC), sustained (SUS), and transient (TRA). Signaling dynamics obtained from D1-trained aiSEGcell (cyan) were qualitatively similar to manually curated nuclear marker derived signaling dynamics (black) [25]. Exemplary signaling dynamics from all independent experiments of D2 are displayed with ground truth response types. The length of signaling dynamics varies due to different experimental settings in D2 experiments. Euclidean distance (D) illustrates similarity between ground truth and aiSEGcell signaling dynamics. Larger D correspond to larger dissimilarity between signaling dynamics and D = 0 for identical signaling dynamics. (c) Single cell signaling dynamics response type assignments were similar between aiSEGcell-derived and fluorescent nuclear marker-derived signaling dynamics. Four aiSEGcell signaling dynamics were classified as outliers due to many missing segmentation masks and were omitted from the confusion matrix for clarity (n = 458 signaling dynamics, N = 5 experiments).

dynamics (NON: 94%, TRA: 36%, INT: 69%, SUS: 88%; Fig 3c). TRA signaling dynamics had a lower classification overlap. However, closer inspection of their dynamics data showed that 'misclassified' aiSEGcell signaling dynamics curves were actually very similar to those from manually curated nucleus segmentations. They had a good average Euclidean distance of 0.51 demonstrating the similarity of aiSEGcell and ground truth signaling dynamics. Thus, the differences in classifying a TRA response do not stem from clear differences in manually curated versus aiSEGcell segmentation, but from the high sensitivity of the used classification parameters to minor changes in curve properties (due to the used cut-offs of signal returning to baseline) [26]. As shown in S5 Fig, these very small segmentation deviations would likely not be considered biologically relevant by an expert.

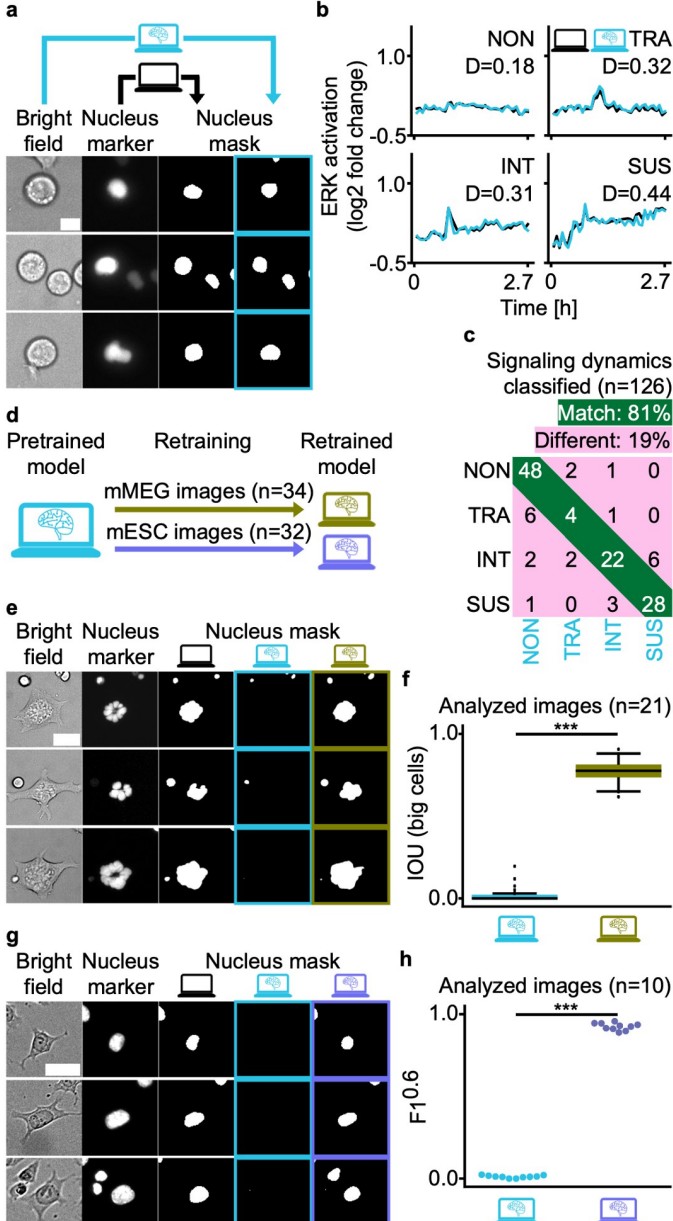

**Fig 3. aiSEGcell accurately segments in unseen experimental conditions and is retrained with little data.** (a) Exemplary segmentations of hMPPs. D1-trained aiSEGcell segmentations (cyan) were qualitatively similar to nuclear marker derived segmentations (black) on unseen experimental conditions. Human MPPs were imaged on a microfluidic chip (scale bar 10 μm). (b) ERK signaling dynamics in hMPPs divide into four response types: non-responsive (NON), transient (TRA), intermediate (INT), and sustained (SUS). Euclidean distance (D) illustrates similarity between manually curated ground truth and aiSEGcell signaling dynamics. (c) Signaling dynamics response type classifications were similar between aiSEGcell and ground truth signaling dynamics (n = 126, N = 1 biological replicate). (d) aiSEGcell trained on D1 was retrained on D3 (n = 34 images, 4,083 cells total, 237 cells with nucleus mask>750 μm$^2$) and D4 (n = 32 images, 3,815 cells) to segment mMEGs (green) and mESCs (purple), respectively. (e-f) Retraining aiSEGcell qualitatively (e) and quantitatively (f) improved nuclear segmentations of mMEGs in BF (scale bar 40 μm). IOU was only computed for large cells (nucleus mask>750 μm$^2$; n = 21 unique field of views, N = 1 biological replicate; two-sided paired Wilcoxon signed-rank test, W = 0, p = 9.5e-7; ***: p<0.001). In boxplots, the central black line is median, box boundaries are upper and lower quartiles, whiskers are 1.5x interquartile range. (g-h) Retraining aiSEGcell qualitatively (g) and quantitatively (h) improved nuclear segmentations of mESCs in BF (scale bar 30 μm; n = 10 unique field of views, N = 1 biological replicate; two-sided paired t-test, degrees of freedom = 9, t = -124.4, p = 7.1e-16). Images were cropped from a larger field-of-view and contrast was individually adjusted here to improve visibility.

Small focus changes due to technical problems or cell differentiation are usually detrimental to segmentations. Therefore, we tested aiSEGcell's robustness against focus drifts. We imaged murine granulocyte/monocyte progenitors (mGMP) with focal planes in the range of ±10 μm (step-size 0.2 μm) equally distributed around the optimal focal plane selected by the experimenter (n = 24 images, N = 1 biological replicate). Segmentations of D1 trained aiSEGcell of all 101 focal planes were compared to the ground truth segmentation in the optimal focal plane. The $F1_{avg}^{0.6}$ was 0.88 versus >0.80 for the optimal focal plane versus a range of 4.4 μm (-1.8 μm to +2.6 μm). This range approximately corresponds to the radius of most primary hematopoietic cells, suggesting that aiSEGcell can accurately segment nuclei in BF as long as the focal plane aligns with parts of the nucleus and demonstrating its robustness to focus changes (S2 and S6 Figs and S7 and S8 Tables).

The diversity of BF imaging data will inevitably lead to use cases to which aiSEGcell trained on D1 does not generalize well. For example, megakaryocytes have large, lobated nuclei that are morphologically very different from the nuclei observed in D1 or D2. Hence, we tested the feasibility of adapting aiSEGcell pretrained on D1 to a data set containing murine megakaryocytes (mMEG) co-cultured with murine hematopoietic stem cells (mHSC; D3; Figs 3d and S7a and S9 and S10 Tables). To prevent overfitting on a single experiment, we used one separate biological replicate for training, validation, and test set, respectively (Table 1). We then retrained aiSEGcell with a small set of 34 images, realistic for a typical experimental laboratory use case, to improve mMEG nucleus segmentations. Whereas the pretrained model could detect nuclei of mHSCs, it failed to segment the much larger nuclei of mMEGs (nucleus mask > 750 μm²; Figs 3e and S2c). Retraining aiSEGcell significantly improved segmentations of mMEG nuclei from an average image-wise IOU of 0.02 to 0.77 (pre- / retrained s.d. = 0.05 / 0.08; Fig 3f and S11 Table). The average image-wise IOU of small cells (i.e. mHSCs; nucleus mask ≤ 750 μm²) improved from 0.38 to 0.64 due to the training data containing both large and small cells (S8 Fig). Following the same strategy, we compiled a data set of adherent non-round murine embryonic stem cells (mESC; D4; Table 1). While the pretrained model failed to segment mESC nuclei, retraining aiSEGcell with 32 images enabled it to segment mESC nuclei in BF images, significantly improving the $F1_{avg}^{0.6}$ from 0.01 to 0.93 (pre- / retrained s.d. = 0.01 / 0.02; Figs 3d, 3g, 3h, S2d, and S7b and S12–14 Tables). Of note, retraining aiSEGcell on D3 or D4 decreased its segmentation accuracy on D2 (D1 trained / D3 retrained / D4 retrained $F1_{avg}^{0.6}$ = 0.66 / 0.27 / 0.12; S3 Table). Training aiSEGcell with only 32 images on a mid-class graphics processing unit (i.e. NVIDIA TITAN RTX) for 1,500 epochs (i.e. iterations over the 32 images) took approximately four hours.

Segmentations with aiSEGcell are based on BF images rich in information about cells, cell interactions, and sub-cellular structures. Understanding which information in BF images is crucial for accurate nucleus segmentations can help efficiently assembling data sets for retraining. Therefore, we selected 999 cells from D2 that aiSEGcell segmented accurately (i.e. high IOU) and 951 cells from D2 that aiSEGcell segmented poorly (i.e. low IOU). Next, we extracted 21 conventional shape and intensity features that describe the nucleus, the whole cell, and the cell neighborhood given the corresponding segmentation masks (S9 and S10 Figs and S15 Table). Taken individually, extracted features were not indicative of whether a cell was segmented accurately or not (S11 Fig). However, training a random forest to classify if a cell is segmented accurately or not (test accuracy = 0.85) revealed that features associated with the nucleus of a cell were more important than features of the whole cell or the cell neighborhood (S12 Fig) [27]. We visually confirmed the importance of nuclear features in BF images by directly interpreting aiSEGcell segmentations with Grad-CAM [28] (S13 Fig).

## Exploring an extra fluorescence channel with aiSEGcell

Removing a fluorescence channel dedicated to image segmentation enables one or even several additional fluorescent markers and thus improved experimental workflows. For example, reporters like GFP are common in transgenic mouse models and are not optimized for multiplexing (i.e. overlapping with cyan and yellow channel). Thus, removing the need for imaging GFP enables the use of both cyan and yellow channels. To illustrate the possibility of analyzing additional biological information in the fluorescent channel freed up by aiSEGcell, we replaced the previously used H2BmCHERRY nuclear marker [25] by virally transducing with fluorescently tagged proliferating cell nuclear antigen (PCNA) to enable live cell cycle phase detection. We isolated lineage unbiased mGMPs (mGMP$^{GM}$) from an existing GFPp65 reporter mouse line and utilized D1-trained aiSEGcell nucleus segmentations to jointly image PCNA, p65, lymphocyte antigen 6 family member C1 (Ly6C), and CD115 (D6; Fig 4a) [29]. We tracked cells for two generations to quantify NfκB signaling dynamics following TNFα stimulation (p65), cell cycle phase states (PCNA), and the differentiation into M versus G lineage committed mGMPs (CD115 versus Ly6C expression; Fig 4b) [30,31]. First, we quantified the distribution of NfκB response types in mother cells. In the control condition, 96% versus 4% of mGMP$^{GM}$ were NON versus TRA (Kull et al.: 96% NON, 0.4% SUS, 4% TRA). In contrast, 81% (15% OSC, 65% TRA) versus 19% of mGMP$^{GM}$ were responsive to TNFα, confirming results from Kull et al. (9% NON, 13% OSC, 2% SUS, 76% TRA; Fig 4c) [25]. Next, we tested if mGMP$^{GM}$ respond differently when stimulated in G1- or S-phase. Irrespective of the cell cycle, TRA responders (49% G1, 57% S) were more frequent than NON (37% G1, 38% S) and OSC responders (14% G1, 5% S; Fig 4d). This trend held true when considering only TNFα stimulated mGMP$^{GM}$, yielding no credible evidence of the cell cycle affecting NfκB responses to TNFα (G1: 18% NON, 18% OSC, 63% TRA, S: 24% NON, 6% OSC, 71% TRA; S14 Fig). Murine GMPs divide into three sub-populations (mGMP$^{GM}$ (Ly6C-CD115-), granulocyte lineage biased mGMPs (Ly6C+CD115-), monocyte lineage biased mGMPs (mGMP$^{M}$; Ly6C +CD115+)) with characteristic NfκB responses that affect cell differentiation [25,31]. Therefore, we quantified the frequency of cell trees that turned on lineage markers Ly6C versus CD115 depending on the NfκB response type. NON cell trees predominantly remained lineage unbiased, whereas OSC and TRA signaling dynamics significantly enriched for Ly6C-CD115 + cell trees (NON / OSC / TRA 87% / 33% / 42% Ly6C-CD115-, 3% / 0% / 0% Ly6C+CD115-, 8% / 58% / 54% Ly6C-CD115+, 3% / 8% / 4% Ly6C+CD115+; Fig 4e). This pattern remained consistent when correcting for the cell cycle, suggesting that mGMP$^{GM}$ primarily differentiate into mGMP$^{M}$ upon TNFα stimulation (Fig 4f). Importantly, these experiments show that aiSEGcell can increase marker multiplexing.

## Using aiSEGcell software

aiSEGcell is an open source software accessible through a CLI (https://github.com/CSDGroup/aisegcell) or a GUI (napari plugin, https://github.com/CSDGroup/napari-aisegcell) [17]. The CLI version can be used to train, test, and predict with aiSEGcell. Training a neural network is challenging and may require tweaking many different parameters for good performance. Therefore, we provide extensive documentation and notebooks to help users getting started with our tool. We asked a group of 5 testers composed of computationally in- and experienced users to install and use the CLI version based only on its online documentation. All testers successfully installed the CLI version within 15 minutes and were able to train, test, and infer with aiSEGcell on macOS, Windows, or Ubuntu. Finally, we provide users with two pretrained models. The model trained on D1 (S1 Fig), and a model trained to segment whole cells in BF.

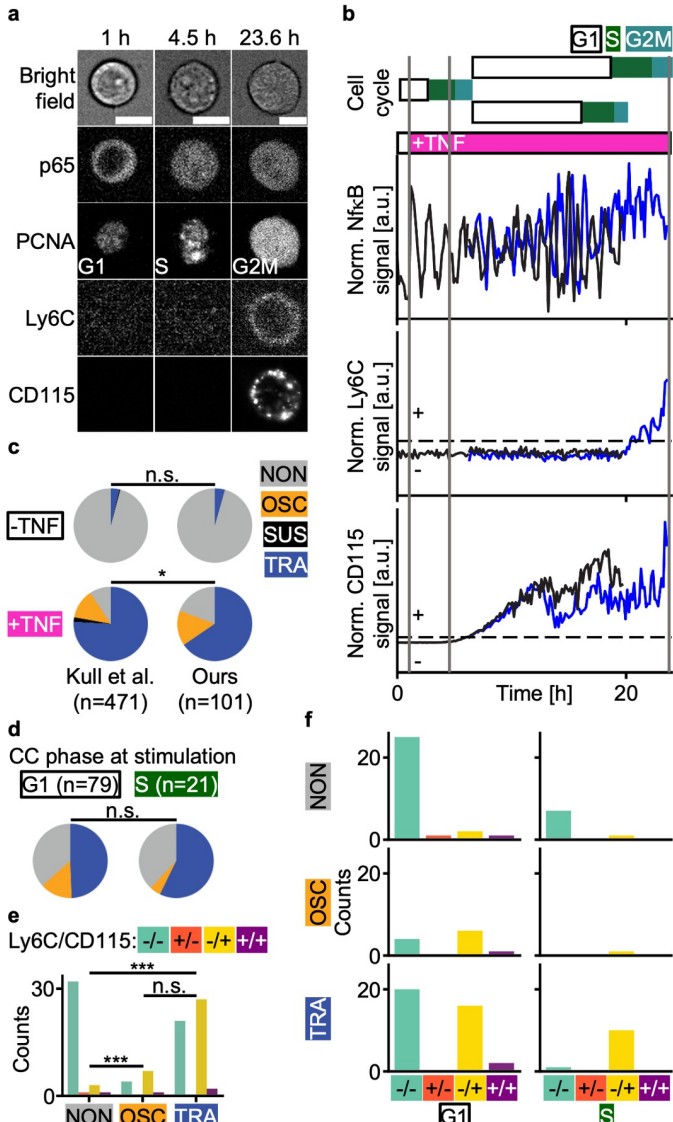

**Fig 4. aiSEGcell permits quantification of an additional fluorescent marker.** Nuclear marker H2BmCHERRY was replaced by aiSEGcell to free the fluorescent channel for cell cycle marker mRUBY2PCNA. (a) BF, p65, PCNA, Ly6C, and CD115 images of a representative cell tree. The cell at 23.6 h is a daughter of the cell at 1 h and 4.5 h (time points as grey vertical lines in (b)). Contrast was individually adjusted here to improve visibility (scale bars 10 μm). (b) Temporal information available for a representative cell tree (n = 101 cell trees, N = 2 biological replicates). Cell cycle phases were derived from PCNA. Mother cell NfκB signaling dynamics were classified into four response types: NON, OSC, SUS, TRA. Cell trees were classified as either positive (+) or negative (-) for Ly6C and CD115, respectively (dashed black lines are threshold). The magenta box illustrates a single TNFα addition after 1 h. Trace of upper daughter cell shown in blue. (c) NfκB response type frequencies were comparable to literature reference in the control/stimulation conditions (-TNF / +TNF; ours n = 23 / 78 cell trees, Kull et al. n = 268 / 203, one-sided Chi$^2$ test, degrees of freedom = 22 / 77, Chi$^2$ statistic = 0.1 / 11.2, p = 0.947 / 0.011; ***: p<0.001, *: p<0.05, n.s.: p≥0.05) [25]. (d) No credible evidence of cell cycle (CC) phase at stimulation affecting NfκB response type frequencies in mGMP$^{GM}$. Signaling dynamics of control and stimulation condition are shown. See S14 Fig for only stimulated cells. One outlier was removed due to weak PCNA signal (one-sided Chi$^2$ test, degrees of freedom = 20, Chi$^2$ statistic = 1.5, p = 0.464). (e) OSC and TRA NfκB responders enriched for Ly6C-CD115+ (-/+) progeny (Ly6C-CD115-: -/-; Ly6C+CD115-: +/-; Ly6C+CD115+: +/+; NON / OSC / TRA n = 37 / 12 / 50; one sided Chi$^2$ test, NON vs. OSC / NON vs. TRA / OSC vs. TRA: degrees of freedom = 11 / 49 / 11, Chi$^2$ statistic = 43.0 / 143.0 / 0.8, p = 2.5e-9 / 8.6e-31 / 0.664). Two outlier cell trees were removed due to confounding Ly6C or CD115 signal. (f) Enrichment of Ly6C-CD115+ cells in OSC and TRA NfκB responders was not mediated by cell cycle phase at stimulation. Three outlier cell trees of (d) and (e) were removed (n = 98 cell trees).

The GUI can only be used for inference and has two modes. The layer mode is intended to explore if representative images of the user's data set can be segmented accurately with available segmentation models or if a new model should be trained with the CLI version. Users can select either one of the two models shipped with the CLI version for segmentation or select a custom model trained in the CLI. If graphical processing units are available, the GUI version will automatically detect and use them enabling faster inference. Moreover, the GUI contains optional postprocessing steps to e.g. dilate or erode masks at run-time. Once the appropriate model and postprocessing settings have been determined in the layer mode, the batch mode enables high-throughput segmentation of images stored even in different folders. While model selection and postprocessing are identical to the layer mode, images are submitted for segmentation via file lists. File lists are agnostic to laboratory specific image storage structure, thus reducing the overhead of copying, renaming and moving imaging data for segmentation. Users can load existing file lists or create new file lists within the GUI. As with the CLI version, we asked a group of six testers with varying computational experience to install and use the GUI version given its online documentation. All testers successfully installed the plugin within 15 minutes and successfully used both modes of the GUI on macOS, Windows, or Ubuntu.

## Discussion

We describe aiSEGcell, a user-friendly software to segment nuclei from only BF images without the need for fluorescent nuclear labeling. Previous research has demonstrated that neural networks can detect nuclei and other cell organelles in transmitted light images under optimized conditions [12,13,15]. We now show that nuclear segmentations in BF are also feasible in challenging common experimental conditions like low magnification, and time lapse imaging of e.g. small non-adherent primary murine and human hematopoietic cells imaged in microfluidic and other culture devices. Notably, we prove that these segmentations are so similar to manual segmentation that they retain biologically meaningful information even when using the segmentation as the basis for quantification of difficult and noisy reporters like biosensors for cell signaling dynamics.

By freeing up a fluorescent channel, aiSEGcell facilitates the exploration of additional fluorescent markers or the reduction of phototoxicity and required imaging times. We demonstrate that aiSEGcell can thus add biological information to an experiment previously constrained due to e.g. poor fluorophore markers in an existing transgenic mouse model. More specifically, we explore the effect of NfκB signaling dynamics on the differentiation of mGMP$^{GM}$ and correct for the cell cycle as a potential covariate. Other signaling pathways like ERK in mESCs change their response dynamics depending on CC phases [23]. In contrast, here we find that cell cycle progression does not affect NfκB signaling responses to TNFα stimulation in mGMP$^{GM}$ cells. At a broader level, we show how aiSEGcell enables new experimental workflows and its utility in jointly investigating multiple signaling pathways, differentiation markers, and control variables.

The ability to generalize to many use cases is a sought-after feature in trained neural networks and it is crucial for democratizing access to state-of-the-art machine learning. Conventionally, large models are trained on vast amounts of data to obtain generalizability [11,32,33]. However, microscopy data are highly diverse and siloed impeding the availability of broadly representative data sets. To address this challenge, crowdsourcing platforms have been used to annotate large data sets [34]. Alternatively, periodic retraining with user provided data has been suggested [20]. Here, we demonstrate that aiSEGcell pretrained on a large data set requires only approximately 30 images to drastically improve segmentation for new use cases.

Consequently, aiSEGcell retraining is feasible for users with little computational resources and improves access to state-of-the-art image segmentation.

Sharing data in FAIR repositories is crucial to advance research on computer vision and disseminate computer vision applications [18]. Hence, we share our data sets D1-5 containing manually curated nuclear marker ground truth segmentations of more than 20,000 images, or 1,100,000 nuclei, covering 11 unique cell types (one cell line, 10 primary cell types; Figs 1–3 and Table 1). With 20,000 images and 1.1 million nuclei, these data sets are among the largest publicly available data sets for fluorescence or label free nucleus segmentation to date [34–37]. Thus, we believe that our data sets can be a valuable resource to the community in advancing image segmentation software.

Programming experience is a major limiting factor preventing the broader adoption of machine learning in biology. New machine learning methods are often available as public code repositories that lack the necessary testing to ensure compatibility with different computational environments and are not actively maintained [12,13]. This is further accentuated in notoriously difficult to train neural networks implemented in complex frameworks that are subject to frequent changes [38,39]. Therefore, it is crucial to release scientific software that is well documented and accessible to programming novices [3,5,20,40]. For the first time, we introduce a software for segmenting nuclei in BF images, proven to work across various computational environments and user coding proficiency levels. We release aiSEGcell with already trained models to segment nuclei and whole cells in BF. The software is available as CLI and GUI versions for easy accessibility. Importantly, aiSEGcell utilizes a generic image segmentation approach not limited to BF microscopy [16]. Hence, users can retrain an existing aiSEGcell model or train a new aiSEGcell model to segment other objects in different 2D image modalities, for example fluorescently labeled blood vessels or cell organelles in electron microscopy, etc., further enhancing its utility in a wide range of applications.

## Materials and methods

### Ethical statement

The research presented here complies with all relevant ethical regulations. Animal experiments were approved according to Institutional guidelines of Eidgenössische Technische Hochschule Zürich and Swiss Federal Law by the veterinary office of Canton Basel-Stadt, Switzerland (approval no. 2655). Relevant ethical regulations were followed, according to the guidelines of the local Basel ethics committees (vote 13/2007V, S-112/2010, EKNZ2015/335) or the ethics boards of the canton Zurich (KEK-StV-Nr. 40/14).

### Mice

Previously published transgenic mice were used for D1, D2, D6 and C57BL/6j mice were used for D3 [25,29]. Experiments were conducted with 10-16-week-old (D1, D2, D6) or 8–12-week-old mice (D3) and only after mice were acclimatized for at least 1 week. Mice were housed in improved hygienic conditions in individually ventilated, environmentally enriched cages with 2–5 mice per cage and ad libitum access to standard diet and drinking water. Mice were housed in a humidity (55 ± 10%) and temperature (21 ± 2 °C) controlled room with an inverse 12 h day-night cycle. Animal facility caretakers monitored the general well-being of the mice by daily visual inspections. Mice displaying symptoms of pain and/or distress were euthanized. Mice were randomly selected for experimental conditions and in some cases pooled to minimize biological variability.

## Genotyping

GFPp65 and H2BmCHERRY transgene expression was verified by flow cytometry. Mice were homozygous for GFPp65 (D6; Fig 4) or homozygous for GFPp65 and heterozygous for H2BmCHERRY transgenes (D1-D2; Figs 1 and 2 and S16 Table) [25,29].

## Isolation of murine hematopoietic cells (D1/D2/D6)

Primary cells were isolated as previously described [25,41]. In brief, femora, tibiae, coxae, and vertebrae were extracted, ground in phosphate buffered saline (PBS; 2% fetal calf serum (FCS), 2 mM ethylenediaminetetraacetic acid (EDTA)) on ice and filtered through a 100-µm nylon mesh. Cells were depleted for erythrocytes for 3 min on ice using ammonium-chloride-potassium (ACK) lysing buffer (Gibco) and stained with biotinylated anti-CD3ε, -CD19, -NK1.1, and -Ly-6G (murine monocytic progenitors) or biotinylated anti-B220, -CD3ε, -CD11b, -CD19, -Ly-6G, and -TER-119. For D6 experiments, biotinylated anti-CD41 was also included in the staining. Streptavidin-conjugated magnetic beads (Roti-MagBeads, Roche) were added for immuno-magnetic separation. GMP subpopulations (CD16/32-PerCy-Cy5.5, CD115-BV421, Sca1-BV510, streptavidin-BV570, cKit-BV711, CD34-eFL660 and Ly6C-APCfire), multipotent progenitor populations (CD135-PerCp-eFl710, Sca1-PacBlue, cKit-BV510, CD150-BV650, streptavidin-BV711, CD34-eFluor660 and CD48-APCeFl780), pre-granulocyte/monocyte progenitors (mPreGM), megakaryocyte progenitors, pre-megakaryocytic/erythroid progenitors (CD41-PerCp-eFl710, CD105-eFl450, Sca1-BV510, streptavidin-BV570, CD150-BV650, cKit-BV711 and CD16/32-APC-Cy7), monocytic progenitors (streptavidin-BV650, CD115-BV-421, cKit-BV711, CD135-APC, CD11b-BV605, Ly6C-BV510), mGMP, and mHSC (CD16/32-PerCp-Cy5.5, Sca1-PacBlue, cKit-BV510, CD150-BV650, streptavidin-BV711, CD34-e-Fluor660 and CD48-APCeFl780) were stained for 90 min on ice, and sorted in single cell mode and sorting purities ≥ 98% with a 70-µm nozzle, using a BD FACS Aria I or III. For D6 experiments, mPreGMs were stained (streptavidin-BV570, CD16/32-PerCy-Cy5.5, CD115-BV421, cKit-BV510, CD150-BV650, Sca1-BV711, Ly6C-APC, CD105-APC/Fire 750) for 90 min on ice and isolated with a 100-µm nozzle in single cell mode and sorting purities ≥ 98% using a BD FACS Aria III. Murine PreGMs were virally transduced for 36 h, stained with the same panel on ice for 90 min to sort mGMP$^{GM}$ positive for monomeric RUBY2 (mRUBY2) with a 100-µm nozzle in single cell mode and sorting purities ≥ 98% using a BD FACS Aria III (S15 Fig and S17 Table).

## Isolation of hematopoietic cells (D3)

Cells were isolated as previously described [42–44]. Femora, tibiae and vertebrae were dissected and crushed in PBS (2% FCS, 2 mM EDTA) on ice. After filtration through a 100-µm nylon mesh, erythrocytes lysis with ACK lysing buffer, and lineage depletion with EASYSep mouse hematopoietic progenitor cell isolation kit (STEMCELL Technologies), cells were stained with anti-CD34-eFluor450, -Sca1-PerCP-Cy5.5, -CD48-FITC, -cKit-PE-Cy7, -CD135-PE-CF594, -CD150-PE, -CD41-APC, and streptavidin-APC-eFl780 for 90 min on ice. mMEGs and mHSCs were subsequently sorted with a BD FACS Aria III (100-µm nozzle, 4-way purity; S17 Table).

## Virus production and transductions (D6)

Lentiviruses were produced and transduced as previously described [45]. In brief, PCNA was fused to fluorescent protein mRUBY2 and cloned into vesicular stomatitis virus glycoprotein pseudotyped lentivirus (third generation) constructs using the In-Fusion Cloning system

(Takara Bio) [46–48]. The virus was produced in human embryonic kidney 293T cells and titrated using NIH-3T3 fibroblasts. Lentivirus was added at a multiplicity of infection of 50–100 to fluorescence-activated cell sorting (FACS) purified mPreGMs in a round bottom 96-well plate for 36 h (37˚C, 5% $O_2$, and 5% $CO_2$) before purifying transduced cells by FACS.

### Primary cell culture of progenitor cells (D1/D2/D6)

Primary cells of experiments 1–7, 13–15, 18–19, and 27–28 were cultured as previously described [25]. After isolation as described above, cells were seeded in 4-well micro-inserts (ibidi) within a 24-well plate with glass bottom (Greiner) and settled for 30–90 min. Before seeding, the plate was coated with anti-CD43-biotin antibody (10 µg/mL in PBS) for 2 h at room temperature and washed with PBS [49]. After cells had settled, 1 mL of IB20/SI media (IB20 = custom IMDM without riboflavin (Thermo Fisher) supplemented with 20% BIT (STEMCELL Technologies), 50 U/mL penicillin (Gibco), 50 µg/mL streptomycin (Gibco), GlutaMAX (Gibco), 2-mercaptoethanol (50 µM; Gibco); SI = 100 ng/mL murine SCF (Pepro-tech), 10 ng/mL murine IL-3 (Peprotech)) was added per well and cells were cultured at 37˚C, 5% $O_2$, and 5% $CO_2$. For D6, Ly6C-APC (1:1,000) and CD115-BV421 (1:1,000) were added to IB20/SI media (S17 Table) [50,51].

### Murine macrophage and osteoclast differentiation (D1/D2)

Common murine macrophages of experiments 8–12 and 16–17 were obtained as previously described [41]. Common monocytic progenitors (Lin[neg]CD115[pos]CD117[pos]CD135[neg], see above) were cultured in MEM alpha (Gibco) with 10% fetal bovine serum (PAA), 1% penicil-lin/streptomycin, 50 ng/mL M-CSF (Peprotech), 1% GlutaMAX, at pH 6.9. Following a 3-day incubation at 37˚C, 5% $O_2$, and 5% $CO_2$, the cells were stimulated with culture media contain-ing 100 µg/mL ascorbic acid (Sigma-Aldrich), with or without 100 ng/mL Receptor activator of NFκB ligand (RANKL; Miltenyi), to induce osteoclast or monocyte-macrophage lineage, respectively. Differentiation into macrophages was morphologically verified (S17 Table).

### Murine embryonic stem cell culture (D4)

Murine ESCs were cultured as previously described [52]. In brief, mESC lines were passaged every 2 days in serum + leukemia inhibitory factor (LIF) containing medium (DMEM (Thermo Fisher), 2 mM GlutaMAX, 1% NEAA (Thermo Fisher), 1 mM sodium pyruvate (Sigma-Aldrich), 50 µM 2-mercaptoethanol, 10% FCS, 10 ng/mL LIF (Cell Guidance Sys-tems)) on gelatin (0.1%; Sigma-Aldrich). Before experiments, cells were cultured for minimum 6 days in 2iLIF medium (NDiff227 (Takara Bio), 1 µM PD0325901 (Selleckchem), 3 µM CHIR99021 (R&D Systems)) on e-cadherin-coated (STEMCELL Technologies) cell-culture plates. All cell lines were tested negatively for mycoplasma using PCR.

### Confocal time lapse imaging (D1/D2)

Experiments 1–7 and 15 have been published and were acquired as previously described [25]. Experiments 8, 9, 12, 16, and 18 were acquired using Nikon NIS acquisition software on a Nikon A1 confocal microscope with a Nikon A1-DUG-2 detector with a 20x/0.75 CFI Plan Apochromat λ objective. Experiments 10, 11, 13, 14, 17, and 19 were acquired using Nikon NIS acquisition software on a Nikon W1 SoRa Spinning Disk confocal microscope with a Hamamatsu Flash 4.0 or a Hamamatsu ORCA-Fusion camera, 20x/0.75 CFI Plan Apochromat λ objective, and Spectra X (Lumencor) light source for BF. Media conditions for murine monocytic progenitors/macrophages and other primary hematopoietic cells were as described

above. Images were acquired every 6–9 min and movie lengths varied between 12–48 h. After 1 h, the movie was briefly stopped to stimulate cells with control media, TNFα- (Peprotech), lipopolysaccharide- (Sigma-Aldrich), or RANKL- (Miltenyi) supplemented media. All time lapse experiments were conducted at 37˚C, 5% O2, and 5% CO2 (S16 Table).

### Time lapse imaging (D3)

Time lapse experiments were conducted as previously described [22,48]. In brief, mHSCs and mMEGs were co-cultured at 37 ˚C, >98% humidity, 5% O2, and 5% CO2 on a µ-slide (ibidi), coated with 5 µg/mL anti-CD43, in phenol-red-free IMDM media (20% BIT, 100 ng/mL SCF, 100 ng/mL thrombopoietin (R&D Systems), 10 ng/mL IL-3, 1x GlutaMAX, 50 µM 2-mercaptoethanol, 50 U/mL penicillin, 50 µg/mL streptomycin) for 24 h before imaging on a Nikon Eclipse Ti-E equipped with linear-encoded motorized stage, 20x/0.75 CFI Plan Apochromat λ objective, Hamamatsu Orca Flash 4.0 V2, and Spectra X fluorescent light source (Lumencor; S16 Table).

### Time lapse imaging (D4)

First, 3,000 mESCs were seeded per channel of an E-Cadherin coated µ-slide (ibidi) in NDiff227 medium containing doxycycline (Dox; 1 ng/mL; Sigma-Aldrich) and overlaid with silicone oil. After approximately 20 h, cells were washed 5x to remove Dox and the medium was replaced with NDiff227 containing different combinations of the small molecules Dox, trimethoprim (Sigma-Aldrich), and 4-hydroxytamoxifen (Sigma-Aldrich). Imaging was performed on Nikon Eclipse Ti-E microscopes equipped with 10x/0.45 CFI Plan Apochromat λ objective, Hamamatsu Orca Flash 4.0 cameras, and Spectra X fluorescent light source (Lumencor). Images were acquired using custom software [53]. All time lapse experiments were conducted at 37˚C, 5% O2, and 5% CO2. Only a single time point of the time lapse experiment was used for all experiments involving D4 (S16 Table).

### Isolation, imaging, and image quantification of hMPPs (D5)

Experiment 26 on hMPPs (S16 Table) has previously been published [22]. Signaling dynamics derived from aiSEGcell segmentations were processed and quantified with the respective analysis pipeline.

### Confocal time lapse imaging (D6)

Experiments were acquired using Nikon NIS acquisition software on a Nikon W1 SoRa Spinning Disk confocal microscope with a Hamamatsu ORCA-Fusion camera, 20x/0.75 CFI Plan Apochromat λ objective, and Spectra X (Lumencor) light source for BF. Media conditions for mGMP$^{GM}$ were as described above. Images were acquired every 9 min for 36 h. After 1 h, the movie was briefly stopped to stimulate cells with control media or TNFα-supplemented media (40 ng/mL final concentration). All time lapse experiments were conducted at 37˚C, 5% O2, and 5% CO2 (S16 Table).

### Image quantification and analysis (D1/D2/D3/D4/D6)

Images were processed and quantified as previously described [5,25,48,54–59]. In brief, 12-bit or 16-bit images were saved as TIFF-files and linearly transformed to 8-bit PNG-files using channel-specific black- and white-points. Fluorescence channels were background corrected for D3 and D4. Images were segmented and individual cells of D2 and D6 were tracked for subsequent fluorescence channel quantification. Missing values due to missing masks were

mean imputed. For D2 NfκB normalization, mean nuclear NfκB intensities were computationally resampled in intervals of 7 min using linear interpolation to account for the different experimental imaging frequencies. Mean NfκB intensity was divided by the cell-wise average of mean NfκB intensity prior to stimulation and rescaled to a range of 0 and 1 using experiment-wise min-max scaling. For D6 NfκB normalization, mean nuclear NfκB intensity was divided by mean whole cell NfκB intensity. Normalized NfκB signal was divided by the cell-wise average of normalized NfκB signal prior to stimulation to obtain baseline normalized NfκB signal used for subsequent analysis. Cell cycle phases were manually assigned during cell tracking. Ly6C and CD115 status was assigned by thresholding on the whole cell mean intensity divided by the cell-wise average of whole cell mean intensity prior to stimulation. For Ly6C, cell-wise mean and standard deviation prior to stimulation were computed to set *threshold = mean+5 * s.d.*. For CD115, cell-wise mean prior to stimulation were computed to set *threshold = 3 * mean*. Assignments of Ly6C and CD115 status were manually confirmed. NfκB signaling dynamics were normalized and classified in R (4.0.2).

## Image segmentation

Ground truth nuclear masks were obtained by manually curating fastER segmentations [5]. In brief, fastER uses human-in-the-loop region annotations to train a support vector machine that segments candidate regions based on shape and texture features and uses a divide and conquer approach to derive an optimal set of non-overlapping candidate regions. fastER was trained experiment-wise on the respective nuclear marker channel and resulting masks were eroded (settings: dilation -2). fastER accurately segmented most nuclei, but light path differences between fluorescence and BF (e.g. shadow obstructing cell in BF) or, for example, focus shifts of live non-adherent cells led to missing or inaccurate object segmentations. Consequently, inaccuracies in segmentations were manually curated with a custom napari plugin if necessary [17]. aiSEGcell segmentations were obtained by predicting and thresholding (0.5) on a single foreground channel from BF images with the respective trained model and subsequently removing small objects (skimage.morphology.remove_small_objects) and small holes (skimage.morphology.remove_small_holes) with experiment-specific settings. Semantic segmentations were converted to instance segmentations using connected components (skimage.measure.label) [60]. Whole cell segmentation masks for D6 experiments were obtained using the aiSEGcell model to segment whole cells in BF accessible through CLI and GUI. For D5 and D6, aiSEGcell segmentations were manually curated in BF if necessary [55].

## High density experiment (D7)

Murine GMPs were isolated from C57BL/6j mice as described above. 10,000 mGMPs were seeded per well of a micro-Insert 4 Well (ibidi) and were stained with SPY555-DNA (1:1,000; Spirochrome) for 300 min at 37°C, 5% O2, and 5% CO2. The experiment was acquired using custom software [53] on a Nikon Eclipse Ti2 with a Hamamatsu ORCA-Fusion camera with a 20x/0.75 CFI Plan Apochromat λ objective, and Spectra X (Lumencor) light source (S16 Table).

## Focus experiment

Murine GMPs were isolated from a C57BL/6j mouse as described above. Cells were stained with SPY650-DNA (1:1,000; Spirochrome) for 60 min on ice. The experiment was acquired using Nikon NIS acquisition software on a Nikon W1 SoRa Spinning Disk confocal microscope with a Hamamatsu ORCA-Fusion camera with a 20x/0.75 CFI Plan Apochromat λ objective, and Spectra X (Lumencor) light source for BF. The experimenter manually selected

the optimal focal plane and focal planes in the range of ±10 μm (step-size 0.2 μm) equally distributed around the optimal focal plane were acquired for 24 field-of-views. Media conditions for mGMPs were as described above. Images were processed as for D1. Ground truth segmentations were obtained using fastER on the nuclear fluorescence signal of the optimal focal plane, (D1-trained) aiSEGcell segmentations were obtained for all focal planes [5].

## NfκB signaling dynamics classification

NfκB signaling dynamics were classified as NON, OSC, SUS, TRA, or unclear/outlier by the experimenter. The experimenter was blind to the experimental conditions of the randomly displayed signaling dynamics and responder-, OSC-, and SUS-TRA-scores were displayed [25].

## F1 computation

We quantified segmentation performance with two types of F1-scores. The conventional F1-score was implemented as previously described [61]. In brief, object-wise IOUs were computed for each object in a ground truth segmentation mask and the predicted segmentation mask. Ground truth objects with an $IOU > \tau_1$ were considered a true positive (TP). Ground truth objects with no matching predicted object ($IOU \leq \tau_1$) and predicted objects with no matching ground truth object were considered FN and FP, respectively. We then computed $F1 = \frac{2*TP}{2*TP+FP+FN+\varepsilon}$, with $\varepsilon$ a small numerical term to prevent division by 0, for varying thresholds $\tau_1$ (0.5–0.9 in 0.05 increments), Conventional F1-scores were provided in S2, S3, S5, S6, S8, S10, S11, S13, and S14 Tables.

   We observed that most errors were nuclei segmented with only small inaccuracies, which were nevertheless counted as FP and FN. This led the conventional F1-score to underestimate the segmentation quality (S16 Fig). Therefore, we introduced a second threshold ($\tau_2 = 0.1$) to prevent counting inaccurately predicted objects (e.g. IOU = 0.4) as both, FP, and as FN. FPs were objects in the prediction with $IOU \leq \tau_2$ and FNs were objects in the ground truth with $IOU \leq \tau_2$. We introduced the category of inaccurate masks (IA), ground truth objects with exactly one $\tau_2 < IOU \leq \tau_1$ and computed the adapted image-wise $F1 = \frac{2*TP}{2*TP+FP+FN+IA+\varepsilon}$. Given the small size of primary cell nuclei at 20x magnification (many nuclei <100 pixels) and the sensitivity of the F1 computation to object size, we observed $\tau_1 = 0.6$ to be an appropriate threshold and denoted the image-wise average adapted F1-score with $\tau_1 = 0.6$ as $F1_{avg}^{0.6}$ throughout this manuscript [21,61,62]. Analogous to the conventional F1-score, we provided the adapted F1-score for varying $\tau_1$ thresholds (S1, S3, S5–S7, S9, S11, S12, and S14 Tables). Ground truth objects with $\tau_2 < IOU \leq \tau_1$ for more than one predicted object were considered a split. Predicted objects with $\tau_2 < IOU \leq \tau_1$ for more than one ground truth object were considered a merge.

## aiSEGcell performance evaluation

For D2 and D4 we evaluated the performance of aiSEGcell using the F1-score. Due to low magnification imaging of small (primary) cells, segmentation masks were in part <100 pixels. Consequently, object-level performance metrics (i.e. F1-score) were more adequate for performance evaluation and less sensitive to noise level deviations of a few pixels. Images in D3 contained both mHSC and mMEG with mMEG segmentation quality being more relevant to the task. While F1-scores helped assess segmentation errors for the co-culture setting they did not qualify to evaluate the segmentation quality of a subset of cells in an image. Hence, we evaluated the performance of aiSEGcell on mMEG using object-wise IOU. Megakaryocyte nuclei were >2,000 pixels in size and noise level deviations of a few pixels did not impede the explanatory power of object-wise pixel-level performance metrics (i.e. IOU).

## aiSEGcell model selection

U-Net was selected as model architecture due to its wide and robust applicability in biological image segmentation [16,63]. Leaky ReLU and batch normalization were used in convolutional blocks [64,65]. Two convolutional blocks and a maximum pooling layer or bilinear upsampling layer formed the down- and upscaling blocks, respectively. Convolutional kernels (3x3) were padded, and the number of kernels doubled with each downscaling block (capped at 512), starting at 32. The number of kernels halved with each upscaling block. In total, 7 downscaling blocks were used to result in a receptive field of 128x128 pixels for the lowest layer kernels. Model instances were trained on randomly augmented (flipping, rotation) 1,024x1,024-pixel images or random crops of 1,024x1,024 pixels from larger images. For D3 and D4 retraining, random blurring/sharpness augmentations were additionally used to improve training on the very small and less diverse training data. Binary cross entropy was used as loss function.

For D1 training, model instances were trained for 400 epochs with a batch size of 14, the maximum batch size to fit on two NVIDIA TITAN RTX GPUs during training. The weight of the loss function was 1. Model instances were initialized with random weights and three replicates for each learning rate (5e-3, 1e-3, 5e-4, 1e-4) were trained [66]. The best model was selected based on the highest average image-wise F1-score on the validation set. For D3 retraining, model instances were initialized with the weights of the best model instance trained on D1 and were trained for 1,500 epochs with a batch size of 4 to balance generalizability and the small training set size. The learning rate was fixed at 1e-5 and three replicates for each loss weight (1, 2, 3, 4, 5, 6, 7, 8, 9, 10, 20, 50, 100, 150, 200) were trained. The best model was selected based on the highest average image-wise IOU for big cells on the validation set. For D4 retraining, model instances were initialized with the weights of the best model instance trained on D1 and were trained for 1,500 epochs with a batch size of 4. A grid search to find the best model instance considered learning rate (5e-5, 1e-5, 5e-6, 1e-6) and loss weight (1, 2, 3) with three replicates for each hyperparameter combination. The best model was selected based on the highest average image-wise F1-score on the validation set.

## Cellpose and StarDist pretrained model evaluation

We used Cellpose version 3.0.7 and selected "nuclei" as a generalist nucleus segmentation model and "yeast_BF_cp3" as a bright field round object segmentation model [20]. A grid search to find the best post-processing hyperparameters considered diameter (17, 30, 40) and flow threshold (0.2, 0.4, 0.6).

We used StarDist version 0.8.5 and selected available pretrained models "2D_versatile_fluo" and "2D_versatile_he" for testing [19]. A grid search to find the best post-processing hyperparameters considered probability threshold (0.5, 0.8, model-specific default) and non-maximum suppression threshold (0.2, 0.4, model-specific default).

## Shape and intensity feature computation

We selected 3,053 images of D2 that did not cause an error during feature computation (i.e. cell contour identification) due to too close cell masks. In this subset of D2, for all experiments selected the 200 cells with the highest IOU and 200 cells with the lowest IOU. After manual curation 1,950 cells (999 high, 951 low) remained, for which nucleus and whole cell masks were available (aiSEGcell whole cell segmentation). The 21 shape and intensity features were computed as described (S15 Table) [60,67].

### Random forest analysis

The best random forest classifier was selected by a grid search over training set size (50, 100, 200, 300, 400, 500, 600, 700, 800, 1,000), number of estimators (20, 50, 100, 200, 300, 400, 500, 600), and maximum estimator depth (5, 7, 9, 11, 13) with 4-fold cross validation. Training and test sets were balanced, and cells were randomly selected from the 1,950 available cells. The best model had 600 estimators with a maximum estimator depth of 7 and was trained on 200 cells. Feature importances were based on the Gini importance. The scikit-learn implementation of random forest was used [68].

### GradCAM analysis

GradCAM heatmap overlays were obtained using the python grad-cam package [69].

### Data visualization and statistical analysis

All data visualization and statistical analyses were conducted in Python (3.10.9). When appropriate, distributions were tested for normality (Shapiro-Wilk test) to select the statistical test for group comparisons [70].

## Supporting information

**S1 Fig. Hyperparameter optimization for D1.** aiSEGcell instances trained with different learning rates on the D1 training set were evaluated with respect to $F1_{avg}^{0.6}$ on the D1 test set (n = 3,153 images, N = 12 experiments). All other hyperparameters were identical between hyperparameter conditions. For each learning rate three random weight initializations were trained. The instance with the highest $F1_{avg}^{0.6}$ (cyan square) was used for all subsequent experiments.
(TIF)

**S2 Fig. Exemplary aiSEGcell segmentations for D2-D5.** Nuclear segmentations in bright field were qualitatively similar to fluorescent nuclear marker derived segmentations. Images were cropped from a larger field-of-view for visibility and stem from a different independent experiment of (a) D2 (scale bar large images 40 μm, zoom ins 10 μm), (b) D5 (scale bar large image 40 μm, zoom in 10 μm), (c) D3 (scale bars 40 μm), and (d) D4 (scale bar large image 40 μm, zoom in 30 μm). Contrast individually adjusted here to improve visibility. Computers represent ground truth (black), aiSEGcell trained on D1 (cyan), aiSEGcell pretrained on D1 and retrained on D3 (green), and aiSEGcell pretrained on D1 and retrained on D4 (purple).
(TIF)

**S3 Fig. aiSEGcell accurately segments nuclei in higher density images.** Nuclear segmentations in bright field were qualitatively similar to fluorescent nuclear marker derived segmentations. Images were cropped from a larger field-of-view (scale bar large image 20 μm, zoom in 10 μm). Contrast individually adjusted here to improve visibility. Computers represent ground truth (black) and aiSEGcell trained on D1 (cyan). Dead cells are crossed-out in green.
(TIF)

**S4 Fig. Signaling dynamics are accurately classified across D2 experiments.** Signaling dynamics are classified into four response types: non-responsive (NON), oscillatory (OSC), sustained (SUS), and transient (TRA). Signaling dynamics were derived from D1-trained aiSEGcell nuclear segmentations (cyan) and manually curated ground truth segmentations (black). Experiment identifiers are taken from S3 Table (experiment 15 / 16 / 17 / 18 / 19

n = 77 / 48 / 47 / 83 / 203 signaling dynamics).
(TIF)

**S5 Fig. D5 differences in transient signaling dynamics classification from ground truth versus aiSEGcell segmentations are due to overly sensitive automated classification, not segmentation errors.** Manually curated ground truth (black) signaling dynamics of D5 were compared to the corresponding signaling dynamics based on D1-trained aiSEGcell (cyan) segmentations. All shown representative examples were classified as transient (TRA) from ground truth traces. The classification based on aiSEGcell segmentation is shown in cyan in each graph. Ground truth and aiSEGcell based signaling dynamics curves are very similar and would not be manually classified differently by an expert user, except maybe the last shown case. Euclidean distance (D) illustrates the high similarity between aiSEGcell and ground truth signaling dynamics. List of abbreviations: non-responsive (NON), intermediate (INT).
(TIF)

**S6 Fig. aiSEGcell is robust to focus changes.** $F1_{avg}^{0.6}$ (black line) for 24 different field-of-views (N = 1 biological replicate, gray area is 95% confidence interval). Z of 0 corresponds to the optimal imaging plane selected by the experimenter. Z offsets were acquired with a step-size of 0.2 μm. Images from +10 μm, +5 μm, +1.8 μm, 0 μm, -1.4 μm, -5 μm, and -10 μm are shown here. Images were cropped from a larger field-of-view and contrast was individually adjusted here to improve visibility (scale bar 10 μm).
(TIF)

**S7 Fig. Hyperparameter optimization for aiSEGcell retraining.** D1 pretrained aiSEGcell, retrained on (a) D3 (hyperparameter: binary cross entropy loss weights; evaluation: average image-wise intersection over union (IOU)) and (b) D4 (hyperparameters: binary cross entropy loss weights, learning rate; evaluation: $F1_{avg}^{0.6}$). All other hyperparameters were identical between hyperparameter conditions. For each hyperparameter combination three random weight initializations were trained. Models were evaluated on the respective validation set (colored square is best model; n = 10 unique field of views, N = 1 biological replicate).
(TIF)

**S8 Fig. Small nucleus segmentations in bright field images improve after retraining.** Average image-wise small cell (nuclei mask≤750 μm$^2$) intersection over union (IOU) of the model pretrained on D1 (murine macrophages, hematopoietic stem- and progenitor cells; cyan; mean = 0.38, s.d. = 0.06) compared to the same model retrained on D3 (murine megakaryocytes, and hematopoietic stem cells; green; mean = 0.64, s.d. = 0.04). Compared to the analysis of big cells, 8 additional images that exclusively contained small cells were included (n = 29 unique field of views, N = 1 biological replicate; two-sided paired t-test, degrees of freedom = 28, t = -20.8, p = 1.4e-18; ***: p<0.001). In boxplots, the central black line is median, box boundaries are upper and lower quartiles, whiskers are 1.5x interquartile range.
(TIF)

**S9 Fig. Workflow to interpret aiSEGcell nucleus segmentations.** Individual cells were selected from D2 (999 high intersection over union (IOU) cells, 951 low IOU cells). Based on ground truth segmentations 21 features were computed for each cell, that described the morphology or intensity of nucleus, whole cell, or cell neighborhood. The random forest classifier with the best set of hyperparameters was trained (and validated) on 200 cells to distinguish between accurately and poorly segmented cells in a test set of 200 cells. Data sets were balanced and randomly compiled. Feature importance (measured by Gini impurity) was derived from the trained random forest classifier. Images were cropped from a larger field-of-view and

contrast was individually adjusted here to improve visibility (scale bar 40 μm).
(TIF)

**S10 Fig. Intensity and shape features computed for each cell.** For each cell (a) 10 nucleus features and (b) 11 whole cell features were computed. For each feature, an example cell that has a high (High) or a low (Low) expression of the respective feature is shown. Images were cropped from a larger field-of-view and contrast was individually adjusted to improve visibility here for shape-based features only (Area, Solidity, Eccentricity, Extent, Distance to closest nucleus, Multinucleated). Contour of nucleus mask in red, contour of whole cell mask in yellow (scale bars 10 μm).
(TIF)

**S11 Fig. Individual features are not sufficient to explain (in)accurate segmentations.** Distribution of (a) 10 nucleus features and (b) 11 whole cell features for high intersection over union (IOU; High; n = 999) and low IOU (Low, n = 951) segmented cells. For violin plots, whiskers depict the minimum and maximum, the central black line is the mean.
(TIF)

**S12 Fig. Nucleus features are essential for aiSEGcell nucleus segmentations.** Feature importance (measured by Gini impurity; larger value means feature is more important) of the 21 features derived from the trained random forest classifier. Features were computed for the whole cell (yellow) or the nucleus (red).
(TIF)

**S13 Fig. Directly interpreting aiSEGcell segmentations highlights importance of nucleus features in bright field.** Grad-CAM of images depicted (a) in Figs 1c and S2a (scale bar large images 40 μm, zoom ins 10 μm), (b) in Figs 3a and S2b (scale bar large image 40 μm, zoom in 10 μm), (c) in Figs 3e and S2c (scale bar large image 40 μm, zoom in 40 μm), and (d) in Figs 3g and S2d (scale bar large image 40 μm, zoom in 30 μm). For bright field (BF) and nucleus marker images the contrast was individually adjusted to improve visibility and Grad-CAM color maps were overlayed. aiSEGcell was trained on D1 (cyan) or retrained on D3 (green) and D4 (purple) after D1 pretraining. In Grad-CAM color maps, colors indicate weak (blue) and strong (red) contributions of pixels to segmented objects.
(TIF)

**S14 Fig. Cell cycle does not affect the frequency of NfκB response types in TNFα stimulated mGMP$^{GM}$.** Only signaling dynamics of mGMP$^{GM}$ stimulated with TNFα are shown. One outlier cell tree was removed due to weak PCNA signal (n = 77 cell trees, N = 2 biological replicates; one-sided Chi$^2$ test, degrees of freedom = 16, Chi$^2$ statistic = 1.8, p = 0.401; n.s.: p≥0.05). List of abbreviations: cell cycle (CC), non-responsive (NON), oscillatory (OSC), transient (TRA).
(TIF)

**S15 Fig. Gating strategies for FACS sorted cell populations in D6.** Isolation strategy for murine pre-granulocyte/monocyte progenitors (mPreGM, first sort) and murine lineage unbiased granulocyte/monocyte progenitor cells (mGMP$^{GM}$, second sort). List of abbreviations: lineage negative (Lin-), lineage negative cKit positive (LK), murine granulocyte/monocyte progenitor (mGMP), monomeric RUBY2 positive (mRUBY2+).
(TIF)

**S16 Fig. Inaccurate segmentations are poorly represented by conventional F1-score.** (a) Inaccurately predicted segmentations that only partially cover the ground truth segmentation

are very common in reality. Despite what schematic representations often suggest, predictions that cover an area that does not correspond to a ground truth object (= false positive) rarely partially overlap with ground truth segmentations. (b) The introduction of a second threshold $\tau_2$ allows counting this common inaccurate mask error (IA) and prevents counting it as false positive (FP) and false negative (FN) error.
(TIF)

**S1 Table. Adapted F1-scores for the D1 test set.** Scores in cells correspond to average adapted F1 +/- standard deviation (n = 3,153 images, N = 12 experiments) and $\tau_1$ refers to the intersection over union threshold above which predictions are considered true positives (best model per $\tau_1$ in bold). The cyan shaded row corresponds to the model we selected for testing and the cyan square in S1 Fig. List of abbreviations: StarDist (SD), Cellpose (CP), 2D_versatile_fluo model (Fluo), 2D_versatile_he model (HE), nuclei model (nuclei), yeast_BF_cp3 model (yeast BF), learning rate (lr), probability threshold (prob), non-maximum suppression threshold (nms), model-specific default (def), diameter (dia), flow threshold (flow).
(DOCX)

**S2 Table. Conventional F1-scores for the D1 test set.** Scores in cells correspond to average conventional F1 +/- standard deviation (n = 3,153 images, N = 12 experiments) and $\tau_1$ refers to the intersection over union threshold above which predictions are considered true positives (best model per $\tau_1$ in bold). The cyan shaded row corresponds to the model we selected for testing and the cyan square in S1 Fig. List of abbreviations: StarDist (SD), Cellpose (CP), 2D_versatile_fluo model (Fluo), 2D_versatile_he model (HE), nuclei model (nuclei), yeast_BF_cp3 model (yeast BF), learning rate (lr), probability threshold (prob), non-maximum suppression threshold (nms), model-specific default (def), diameter (dia), flow threshold (flow).
(DOCX)

**S3 Table. F1-scores for the D2 test set.** Scores in cells correspond to average conventional or adapted F1 +/- standard deviation (n = 6,243 images, N = 5 experiments) and $\tau_1$ refers to the intersection over union threshold above which predictions are considered true positives. Rows of best models trained on D1 (cyan), retrained on D3 (green), and retrained on D4 (purple) are shaded and correspond to the respectively colored squares in S1 and S7 Figs.
(DOCX)

**S4 Table. Test set performance for all data sets.** The average (avg) refers to the average of image-wise computed performance metrics. The aggregate (agg) of performance metrics refers to a data set-wise computation. Data set D7 refers to the higher density experiments. All metrics were acquired using $\tau_1 = 0.6$ and $\tau_2 = 0.1$. List of abbreviations: true positive (TP), false negative (FN), false positive (FP), intersection over union (IOU).
(DOCX)

**S5 Table. F1-scores for D7 test set.** Scores in cells correspond to average conventional or adapted F1 +/- standard deviation on the higher density experiments data set D7 (n = 10 images, N = 3 experiments) and $\tau_1$ refers to the intersection over union threshold above which predictions are considered true positives. D1 trained model (cyan) corresponds to the respectively colored square in S1 Fig.
(DOCX)

**S6 Table. F1-scores for the D5 test set.** Scores in cells correspond to average conventional or adapted F1 +/- standard deviation (n = 816 images, N = 1 experiment) and $\tau_1$ refers to the intersection over union threshold above which predictions are considered true positives. D1

trained model (cyan) corresponds to the respectively colored square in S1 Fig.
(DOCX)

**S7 Table. Adapted F1-scores for the focus tests.** Scores in cells correspond to average adapted F1 +/- standard deviation (n = 24 images, N = 1 experiment) and $\tau_1$ refers to the intersection over union threshold above which predictions are considered true positives. Analysis corresponds to S6 Fig and rows shaded in grey mark z-layers -1.8 μm to +2.6 μm.
(DOCX)

**S8 Table. Conventional F1-scores for the focus tests.** Scores in cells correspond to average conventional F1 +/- standard deviation (n = 24 images, N = 1 experiment) and $\tau_1$ refers to the intersection over union threshold above which predictions are considered true positives. Analysis corresponds to S6 Fig and rows shaded in grey mark z-layers -1.8 μm to +2.6 μm.
(DOCX)

**S9 Table. Adapted F1-scores for the D3 validation set.** Scores in cells correspond to average adapted F1 +/- standard deviation (n = 10 images, N = 1 experiment) and $\tau_1$ refers to the intersection over union threshold above which predictions are considered true positives (best model per $\tau_1$ in bold). The green shaded row corresponds to the model we selected for testing and the green square in S7a Fig.
(DOCX)

**S10 Table. Conventional F1-scores for the D3 validation set.** Scores in cells correspond to average conventional F1 +/- standard deviation (n = 10 images, N = 1 experiment) and $\tau_1$ refers to the intersection over union threshold above which predictions are considered true positives (best model per $\tau_1$ in bold). The green shaded row corresponds to the model we selected for testing and the green square in S7a Fig.
(DOCX)

**S11 Table. F1-scores for the D3 test set.** Scores in cells correspond to average conventional or adapted F1 +/- standard deviation (n = 29 images, N = 1 experiment) and $\tau_1$ refers to the intersection over union threshold above which predictions are considered true positives. Rows of best models trained on D1 (cyan) and retrained on D3 (green) are shaded and correspond to the respectively colored squares in S1 and S7a Figs.
(DOCX)

**S12 Table. Adapted F1-scores for the D4 validation set.** Scores in cells correspond to average adapted F1 +/- standard deviation (n = 10 images, N = 1 experiment) and $\tau_1$ refers to the intersection over union threshold above which predictions are considered true positives (best model per $\tau_1$ in bold). The purple shaded row corresponds to the model we selected for testing and the purple square in S7b Fig. List of abbreviations: learning rate (Lr).
(DOCX)

**S13 Table. Conventional F1-scores for the D4 validation set.** Scores in cells correspond to average conventional F1 +/- standard deviation (n = 10 images, N = 1 experiment) and $\tau_1$ refers to the intersection over union threshold above which predictions are considered true positives (best model per $\tau_1$ in bold). The purple shaded row corresponds to the model we selected for testing and the purple square in S7b Fig. List of abbreviations: learning rate (Lr).
(DOCX)

**S14 Table. F1-scores for the D4 test set.** Scores in cells correspond to average conventional or adapted F1 +/- standard deviation (n = 10 images, N = 1 experiment) and $\tau_1$ refers to the intersection over union threshold above which predictions are considered true positives. Rows of

best models trained on D1 (cyan) and retrained on D4 (purple) are shaded and correspond to the respectively colored squares in S1 and S7b Figs.
(DOCX)

**S15 Table. Morphology and intensity features.**
(DOCX)

**S16 Table. List of experiments and experimental specifications.**
(XLSX)

**S17 Table. List of Antibodies.**
(DOCX)

## Acknowledgments

We thank the D-BSSE single cell facilities for technical support, Arne Wehling for discussions about aiSEGcell, and Linus Angenendt, Germán Camargo Ortega, Aron Wallace Toon Kirschner, Benian Uzun, Aaron Ponti, and Kevin Akira Yamauchi for testing the usability of aiSEGcell.

## Author Contributions

**Conceptualization:** Daniel Schirmacher, Timm Schroeder.

**Data curation:** Daniel Schirmacher, Ümmünur Armagan.

**Formal analysis:** Daniel Schirmacher, Ümmünur Armagan.

**Funding acquisition:** Daniel Schirmacher, Timm Schroeder.

**Investigation:** Daniel Schirmacher.

**Methodology:** Daniel Schirmacher, Ümmünur Armagan.

**Project administration:** Timm Schroeder.

**Resources:** Daniel Schirmacher, Yang Zhang, Tobias Kull, Markus Auler.

**Software:** Daniel Schirmacher.

**Supervision:** Timm Schroeder.

**Validation:** Ümmünur Armagan, Yang Zhang, Tobias Kull, Markus Auler, Timm Schroeder.

**Visualization:** Daniel Schirmacher, Ümmünur Armagan, Timm Schroeder.

**Writing – original draft:** Daniel Schirmacher, Yang Zhang, Timm Schroeder.

**Writing – review & editing:** Daniel Schirmacher, Yang Zhang, Timm Schroeder.

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
