## [Decision Letter · Decision Letter 0]

9 Apr 2024

Dear Prof. Dr. Schroeder,

Thank you very much for submitting your manuscript "aiSEGcell: user-friendly deep learning-based segmentation of nuclei in transmitted light images" for consideration at PLOS Computational Biology.

As with all papers reviewed by the journal, your manuscript was reviewed by members of the editorial board and by several independent reviewers. In light of the reviews (below this email), we would like to invite the resubmission of a significantly-revised version that takes into account the reviewers' comments.

Dear Professor Schroeder,

Thank you for submitting your work to PLOS Computational Biology. The two referees that have looked at your manuscript agree that the paper presents valuable contributions for the community. However, they also agree that there are some issues that need to be address. Specifically, two major issues need your attention in a revised version of the manuscript: 1) the evaluation metric used should be consistent with previous literature, and should be expanded in detail across datasets and other choices. 2) A comparative evaluation with existing methods such as Cellpose or microSAM would be necessary to understand benefits and challenges of the proposed software.

Please, see the details of the reviewer's comments below. Please, let me know if you have any questions.

Best,

Juan C. Caicedo

We cannot make any decision about publication until we have seen the revised manuscript and your response to the reviewers' comments. Your revised manuscript is also likely to be sent to reviewers for further evaluation.

Sincerely,

Juan Caicedo

Guest Editor

PLOS Computational Biology

Daniel Beard

Section Editor

PLOS Computational Biology

Dear Professor Schroeder,

Thank you for submitting your work to PLOS Computational Biology. The two referees that have looked at your manuscript agree that the paper presents valuable contributions for the community. However, they also agree that there are some issues that need to be address. Specifically, two major issues need your attention in a revised version of the manuscript: 1) the evaluation metric used should be consistent with previous literature, and should be expanded in detail across datasets and other choices. 2) A comparative evaluation with existing methods such as Cellpose or microSAM would be necessary to understand benefits and challenges of the proposed software.

Please, see the details of the reviewer's comments below. Please, let me know if you have any questions.

Best,

Juan C. Caicedo

Reviewer's Responses to Questions

**Comments to the Authors:**

Reviewer #1: The paper (will) provide two assets:

The dataset of brightfield nuclei (1.1M nuclei - 20K images).

A model trained on a subset of this dataset (+napari plugin \\ starter code) - already available, repository is well-documented.

The paper does not propose anything novel in terms of computational methods (training of basic U-Net with in-house dataset) nor does it have interesting choices in the pipelines, besides some questions arise with regard to the evaluation metric and experimental side.

While training on Dataset 1 (D1), why no brightness augmentations were used during training?

How does performance of the model change on the D1 dataset after fine-tuning on D3\\D4?

Evaluation metric: authors use unconventional F1-score with an unconventional fixed threshold (which is not motivated either) and seems like it was completely different to the cited source (where multiple IoU thresholds are used). I would recommend here to fall back to more conventional metrics (or even reconsider the choice of metrics). Two papers came out in Nature Methods that might help you:

https://www.nature.com/articles/s41592-023-01942-8

https://www.nature.com/articles/s41592-023-02150-0

Did you consider any existing pretrained model to compare your model with?

On a positive side, authors evaluate segmentation w.r.t. downstream biological tasks by training random forest classifiers with morphological features (21) that were extracted from segmented regions.

In figure 1d: would be nice to have a threshold (0.6) on the plot and also mean or median value.

Reviewer #2: The manuscript describes a new tool for nucleus segmentation in bright field microscopy. It is based on the established U-Net architecture and a large dataset that is collected by the authors for this purpose. The manuscript first evaluates the performance of the segmentation network on (a train/test split of) this dataset, then evaluates the performance for a downstream biological application, investigates the transfer to different bright field datasets, demonstrates the utility of the method by using the now available fluorescent channel for different readouts and finally describes the tool that makes the method available via CLI and napari.

Overall the manuscript is well written and the description of methods and experiments is sound. The software seems to be well documented and relatively easy to use (I did not have time to closely review it).

A reliable method for nucleus segmentation in bright field images is a valuable contribution.

However, two points should be addressed before accepting the manuscript:

1. The method description for the nucleus segmentation is not sufficient. While the authors describe the U-Net architecture they do not describe at all how an instance segmentation is obtained from the predictions of the U-Net.

My best guess given the description given is that the U-Net only predicts a single channel for foreground and then thresholding and connected components are applied. If this is indeed the case, then this work has a major drawback: it will likely not work well for images with a high-confluency where nuclei can be almost touching and will thus be likely merged incorrectly into a single object. (And this fact is the reason for the complex post-processing procedures in StarDist, CellPose, etc.).

To address this weakness:

- Please describe the post-processing of the network outputs in detail.

- If my assumption about the simple post-processing is correct please address the case of images with high confluency:

-- Are such images part of the training / testing data? Did you explicitly check for incorrect merges for images with high confluency?

-- If this is indeed a problem: I would suggest to address this issue by including a slightly more complex network objective and post-processing, e.g. predicting both foreground and boundaries between nuclei, and adapting the post-processing, to enable separation of close or touching objects.

2. I find the presentation of results for the retraining on new datasets confusing. Figure 3 mixes evaluation of the downstream analysis with qualitative results and a quantitative evaluation of the F1-Score and does so inconsistently across the different new datasets. I would suggest to create a figure or table that compares the F1-Score of your model before and after retraining for all new datasets in a consistent manner. This can either be added to Figure 3 (potentially replacing f and h since this information should be in the new element) or be added to the supplementary material.

Besides these two points I have a few minor comments and suggestions.

- A comparison to CellPose would be helpful to understand the practical impact the method has over existing methods. For this you could compare application to one of the new datasets, ideally even with fine-tuning of CellPose. (Such experiments always have many degrees of freedom, so I will not make a recommendation for acceptance dependend on it. Nevertheless, I think an accurately reported comparison would be helpful for the the reader.)

- The FastER used for segmentation of the fluorescence channel could be briefly explained in "Materials: Image Segmentation". How does it work, how reliable was it (qualitatively).

- Since the data is not yet available I could not review the quality or documentation of the dataset. Since this dataset is one of the main contributions of this publication that would have been desirable. However, the authors give a clear plan on how they will make the data available, so I will not make my recommendation available on it. As a sidenote:is there a particular reason to not go for a more centralized solution like BioImageArchive (or similar EBI repository) or zenodo? I personally think these solutions are better suited to FAIR data access than institute specific solutions. (To be clear, this is not relevant for my recommendation.)

**Have the authors made all data and (if applicable) computational code underlying the findings in their manuscript fully available?**

Reviewer #1: **No: **Dataset is not yet available, code is available.

Reviewer #2: **No: **See comments on the dataset in the main review.

PLOS authors have the option to publish the peer review history of their article (what does this mean?). If published, this will include your full peer review and any attached files.

Reviewer #1: No

Reviewer #2: **Yes: **Constantin Pape
---

## [Decision Letter · Decision Letter 1]

24 Jul 2024

Dear Prof. Dr. Schroeder,

We are pleased to inform you that your manuscript 'aiSEGcell: user-friendly deep learning-based segmentation of nuclei in transmitted light images' has been provisionally accepted for publication in PLOS Computational Biology.

Best regards,

Juan Caicedo

Guest Editor

PLOS Computational Biology

Daniel Beard

Section Editor

PLOS Computational Biology

The new version of the paper has addressed the comments raised by the reviewers. We are glad to accept this new version for publication.

However, please note the feedback provided by Reviewer #2 and incorporate a "Limitations" section that discusses the weaknesses pointed out. Make sure to also clarify in that section that the baseline models were not finetuned, and how the results may change by doing so.

Reviewer's Responses to Questions

**Comments to the Authors:**

Reviewer #1: Overall, I accept the answers of the authors. Good to see that the manuscript was improved.

I would only remove results of CellPose`2D_versatile_he model` as it is very different data domain and no surprise the scores are slightly above 0, therefore those results don't add any value.

Reviewer #2: The authors address the points brought up in my review. In particular they now describe the post-processing procedure for instance segmentation, perform an additional experiment to judge the utility of the method for high-density cells and compare to other pre-trained methods.

I see one remaining conceptual weakness with the approach and experiments: the simplistic post-processing employed here means that instance segmentation will inevitably fail for touching objects. Using only foreground based segmentation cannot address such scenario by design. While nuclei may not touch as frequently in the BF images, this absolutely does happen for crowded cells. I have worked with similar scenarios of crowded HeLA and IPSC cells where the nuclei cover the majority of the cytosol visible in BF / Phase-Contrast and the segmentation approach used here would fail. I want to highlight again that the main reason for more sophisticated instance segmentation approaches, from predicting an additional boundary channel to distance based segmentation objectives as in StarDist and CellPose, is to circumvent this problem. The fact that this does not show up in the high-density experiments here just means that the cell density is not high enough yet for touching objects to be a major problem. Another (minor) weakness is the limited comparison to StarDist and CellPose. The authors only apply the pretrained models, which predictably fail. It would have been more illuminating to retrain one of these methods on the dataset here.

Despite these weaknesses, this is an overall solid contribution, especially due to the dataset and experiments on how to use the "freed up" channel, which I think can be accepted. However, I think it would be warranted to add a limitation section that points out that the segmentation methods by design cannot deal with touching objects and that this means that it is not applicable if touching nuclei are prevalent in the data to be analyzed.

**Have the authors made all data and (if applicable) computational code underlying the findings in their manuscript fully available?**

Reviewer #1: Yes

Reviewer #2: Yes

PLOS authors have the option to publish the peer review history of their article (what does this mean?). If published, this will include your full peer review and any attached files.

Reviewer #1: No

Reviewer #2: **Yes: **Constantin Pape

---

## [Editor Report · Acceptance letter]

8 Aug 2024

PCOMPBIOL-D-24-00190R1 

aiSEGcell: user-friendly deep learning-based segmentation of nuclei in transmitted light images

Dear Dr Schroeder,

I am pleased to inform you that your manuscript has been formally accepted for publication in PLOS Computational Biology. Your manuscript is now with our production department and you will be notified of the publication date in due course.

With kind regards,

Anita Estes
